# INFOSYNTH: INFORMATION-GUIDED BENCHMARK SYNTHESIS FOR LLMS

## ABSTRACT

Large language models (LLMs) have demonstrated significant advancements in reasoning and code generation. However, efficiently creating new benchmarks to evaluate these capabilities remains a challenge. Traditional benchmark creation relies on manual human effort, a process that is both expensive and time-consuming. Furthermore, existing benchmarks often contaminate LLM training data, necessitating novel and diverse benchmarks to accurately assess their genuine capabilities. This work introduces InfoSynth, a novel framework for automatically generating and evaluating reasoning benchmarks guided by information-theoretic principles. We propose metrics based on KL-divergence and entropy to quantify benchmark novelty and diversity without relying on costly model evaluations. Building on this framework, we develop an end-to-end pipeline that synthesizes robust Python coding problems from seed datasets using genetic algorithms and iterative code feedback. Our method generates accurate test cases and solutions to new problems 97% of the time, and the synthesized benchmarks consistently exhibit higher novelty and diversity compared to their seed datasets. Moreover, our algorithm provides a method for controlling the novelty/diversity and difficulty of generated problems. InfoSynth offers a scalable, self-verifying pipeline for constructing high-quality, novel and diverse benchmarks for LLMs.

## 1 INTRODUCTION

Large language models (LLMs) have demonstrated impressive capabilities in code generation and reasoning. However, rigorously evaluating these reasoning abilities remains a significant challenge. While substantial effort has been invested in creating robust math and coding benchmarks (Austin et al., 2021; Chen et al., 2021; Cobbe et al., 2021; Jain et al., 2024; Liu et al., 2024a; Zhuo et al., 2024), their development often demands considerable human labor or extensive computational resources for problem and solution validation. Some existing approaches utilize a judge LLM to generate and verify new problems (Ding et al., 2024; Li et al., 2025b; Majumdar et al., 2024). However, this method can yield erroneous benchmarks, as the judge LLM may not reliably solve the generated problems. This paper focuses on Python coding problems, whose solutions can be verified by executing them in a code environment. Our novel pipeline leverages this executability to ensure the robustness of the generated problems.

Beyond the challenge of ensuring robustness, state-of-the-art (SOTA) reasoning models often overfit to their training data, leading to poor performance on out-of-distribution problems (Huang et al., 2025). Furthermore, recent studies have revealed that LLM training data is frequently contaminated by existing evaluation benchmarks, which can artificially inflate reported performance (Deng et al., 2024a;b; Golchin & Surdeanu, 2023). For instance, Zhang et al. (2024) show that LLMs experience accuracy drops of up to 8% on their novel GSM1k dataset, despite its similarity in difficulty to the widely used GSM8k. This underscores the critical need for new, contamination-free reasoning benchmarks to genuinely assess the capabilities of LLMs.

To address these pressing issues, our work emphasizes two crucial benchmark properties: *novelty* and *diversity*. While this work does not directly address the task of creating contamination-free benchmarks, we provide an improved method of generating benchmarks that cover more diverse and novel coding tasks. A novel benchmark should comprise problems distinct from existing datasets, thereby preventing models from achieving high scores through mere memorization of previously

seen examples. Conversely, a diverse benchmark should encompass a broad spectrum of dissimilar problems, enhancing its resilience against model overfitting and providing a more comprehensive evaluation. Clearly, robust, novel, and diverse benchmarks are essential for the reliable evaluation of LLM reasoning abilities. Our work seeks to answer two fundamental questions: (1) How can we effectively measure the novelty and diversity of benchmarks? (2) How can we efficiently generate benchmarks that possess these desirable properties while ensuring their correctness and robustness?

Our main contributions can be summarized as:

- We introduce an information-theoretic framework to quantify and compare the *novelty* and *diversity* of benchmarks, offering a principled approach to benchmark assessment without reliance on model evaluations.
- We propose and validate an end-to-end pipeline, InfoSynth, for efficiently synthesizing novel, diverse, and verifiably correct Python coding problems from seed datasets with genetic algorithms and iterative code feedback.
- Through extensive experiments, we demonstrate that InfoSynth exhibits superior robustness compared to existing data generation methods. Our pipeline provides a method for increasing the novelty and diversity of generated problems and controlling their difficulty.

## 2 RELATED WORK

**Synthetic Problem Generation.** Previous work has explored generating novel synthetic datasets from high-quality seed benchmarks. Majumdar et al. (2024); Wang et al. (2023); Xu et al. (2024); Zhao et al. (2025) show that LLMs can generate new instructions from existing ones. Chen et al. (2023); Liu et al. (2023; 2024b); Xu et al. (2025); Zeng et al. (2025) successfully used LLMs to generate unit tests for solution verification. Our end-to-end pipeline extends existing methods with code-execution environments to ensure robustness, novelty, and diversity.

**Benchmark Quality Assessment.** Efficient, concrete analysis of benchmark quality remains an open problem. Prior work defines metrics for novelty, separability, and difficulty via test-taker performance (Li et al., 2025a;b), proposes adaptive selection of novel problems to reduce evaluation cost (Truong et al., 2025), and develops similarity and difficulty scores for coding tasks (Tambon et al., 2024). A key limitation is reliance on SOTA model performance, making these methods accurate but computationally expensive. Our approach analyzes novelty and diversity more efficiently, without requiring costly model evaluations.

## 3 DESIRABLE BENCHMARK PROPERTIES

We propose a framework for characterizing the novelty and diversity of benchmarks. Our new novelty metric uses the KL-Divergence to capture how different the benchmark is from existing datasets, with the broader goal of creating benchmarks that are contamination-free. Previous work has used the KL-Divergence in a similar way; Schulman et al. (2017) use it to measure differences in reinforcement learning policies and Kingma & Welling (2014) use it to regularize output distributions for Variational Autoencoders. Similarly, our proposed diversity metric uses Shannon entropy to capture how much variety exists among the problems; more diverse datasets provide a broader characterization of an LLM's reasoning abilities.

### 3.1 AN INFORMATION-THEORY BASED FRAMEWORK FOR BENCHMARK ANALYSIS

Formally, a baseline dataset can be modeled as samples $X = \{x_i\} \subseteq \mathbb{R}^d$ drawn from some true distribution $p(x)$, and the new dataset that we want to compare against the baseline can be modeled as samples $Y = \{y_i\} \subseteq \mathbb{R}^d$ drawn from a distribution $q(x)$. Here, $x_i, y_i$ represent the embedding vectors of the problem statements in an embedding space $\mathbb{R}^d$. We define the **novelty** of the new dataset $Y$ to be the KL-divergence between the distributions

$$\text{Novelty}(Y|X) = D_{KL}(q \,\|\, p) = \int_{\mathbb{R}^d} q(\mathbf{x}) \log \frac{q(\mathbf{x})}{p(\mathbf{x})} \, d\mathbf{x} \tag{1}$$

Note that we take the KL-divergence of $p$ with $q$ as the null hypothesis because we want to reward datasets where $q(x)$ is large and $p(x)$ is small, indicating that the dataset contains problems not in

the distribution of the seed dataset. Given a dataset $X = \{x_i\}, x_i \sim p(x)$, we define the **diversity** of the dataset to be the differential entropy of its distribution.

$$\text{Diversity}(X) = -\int_{\mathbb{R}^d} p(\mathbf{x}) \log p(\mathbf{x}) \, d\mathbf{x} \tag{2}$$

Intuitively, the KL-Divergence captures the fact that novel datasets should have different embeddings from existing datasets. Similarly, diverse datasets should have embeddings that are fairly spread out, as clusters indicate problems that are likely to be similar and not diverse. A "perfectly diverse" dataset should resemble a uniform distribution over the embedding space so that it covers a large class of problems. Since the uniform distribution maximizes entropy, our metric captures this intuitive characterization of diversity.

In practice, obtaining the full distribution of the embedding space is intractable. Instead, we use statistical estimators for the KL-Divergence and differential entropy. Given samples $x_1, ..., x_n, y_1, ..., y_m \in \mathbb{R}^d$ where $x$ and $y$ are drawn from $p(x), q(x)$ respectively, we use the k-NN based estimator by Wang et al. (2009)

$$D_{KL}(q||p) = \frac{d}{m} \sum_{i=1}^{m} \log \frac{\nu_k(i)}{\rho_k(i)} + \log \frac{n}{m-1} \tag{3}$$

where $\nu_k(i)$ is the distance from $y_i$ to its $k$-th nearest neighbor in $\{x_j\}$ and $\rho_k(i)$ is the distance from $y_i$ to its $k$-th nearest neighbor in $\{y_j \mid j \neq i\}$; $k$ is a hyperparameter.

Similarly, we can estimate the differential entropy of a dataset. Given samples $x_1, ..., x_N \in \mathbb{R}^d$, the Kozachenko-Leonenko estimator is

$$h(X) = \psi(N) - \psi(k) + \log V_d + \frac{d}{N} \sum_{i=1}^{N} \log \rho_k(i) \tag{4}$$

where $\psi$ is the digamma function, $V_d$ is the volume of the unit ball in $\mathbb{R}^d$, and $\rho_k(i)$ is the distance between $x_i$ and its $k$-th nearest neighbor in $\{x_j \mid j \neq i\}$; $k$ is a hyperparameter.

Computing embeddings, KL-divergence, and entropy for a text dataset is significantly faster and cheaper than computing test-taker statistics. Hence, we provide a way to cheaply estimate the quality of new benchmarks. Moreover, using these metrics, algorithm development can be formulated as an optimization problem that tries to maximize the novelty and diversity of the new dataset.

## 3.2 EMPIRICAL VALIDATION

In this section, we empirically verify the correctness of our metrics on existing datasets. We use all-mpnet-base-v2 (Song et al., 2020) to embed the questions in $\mathbb{R}^{768}$. Because estimating entropy in high dimensions is difficult due to the curse of dimensionality, we project down to a lower dimension using UMAP (McInnes et al., 2018). Note that any two datasets that we want to compare must be projected down in the same UMAP call, preserving their relative geometry. We renormalize embeddings after projection so that distances between embeddings corresponds to cosine similarity.

### 3.2.1 KL-DIVERGENCE METRIC VALIDATION

We use a dataset of 3511 Leetcode problems with concept labels for every problem (kaysss, 2025). We extract three smaller datasets from this: problems tagged "Hash Map" (686 problems), problems tagged "Graph" (160 problems), and problems tagged "String" (786 problems). We also use the MBPP test dataset (374 problems) Austin et al. (2021). We also use HumanEval (164 problems) and 500 randomly chosen problems from the APPS dataset Chen et al. (2021); Hendrycks et al. (2021). We run UMAP with 80 nearest neighbors and a minimum distance of 0.1; we compute this over 10 independent UMAP runs using $k = 4$ for the novelty k-NN parameter.

As shown in Figure 1, our results align with intuition, confirming the KL divergence as a measure of benchmark novelty. In the second graph, the estimator becomes negative despite KL being theoretically nonnegative. This is because we are comparing a subset against a superset: subset–superset distances are small, but intra-subset distances are large, causing the estimator to be negative. We find that in practice, for practically useful comparisons, negativity never occurs; we include this case primarily to illustrate that our metric still reflects human intuition. Note that our comparisons can still work in such degenerate cases as we only care about *relative* differences between datasets.

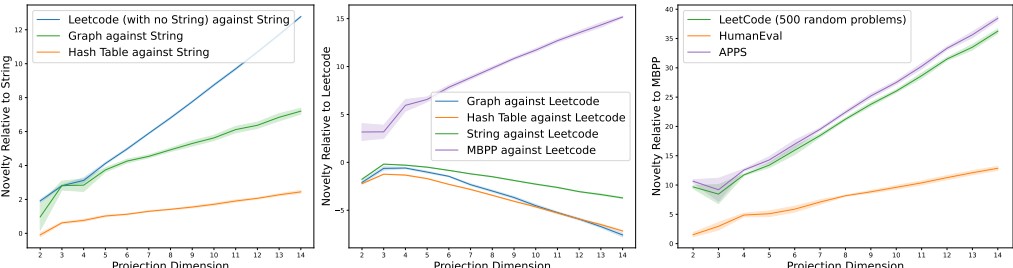

Figure 1: Left: The full Leetcode dataset has higher novelty than its Hash Table and String subsets as expected. Middle: The MBPP dataset has high novelty against the Leetcode dataset, whereas the Leetcode subdatasets have very little relative novelty as expected. Right: Leetcode and APPS have high novelty as their problems are harder and very different from MBPP, whereas HumanEval has low novelty as it is known to be more similar to MBPP. All plots show 95% confidence intervals.

### 3.2.2 DIFFERENTIAL ENTROPY METRIC VALIDATION

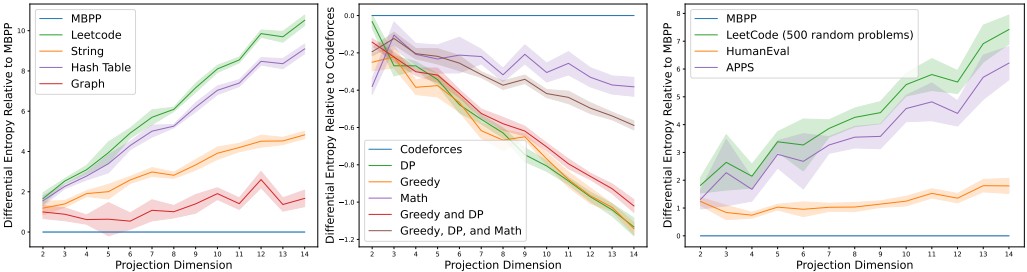

Figure 2: Left: Leetcode vs. MBPP entropy. MBPP shows lower entropy due to simpler, more repetitive problems. Middle: Codeforces vs. subsets. Full datasets have higher entropy than topic-specific subsets, except Math, which overlaps with others (e.g., DP, Greedy) and thus appears highly dispersed when isolated. Right: Leetcode and APPS have high diversity as their problems span many different computer science topics, whereas HumanEval has low diversity as the problems are easier and known to be more similar to MBPP. All plots show 95% confidence intervals.

In addition to previous datasets, we analyze 4,000 random Codeforces problems (open-r1, 2024). UMAP is run with 80 neighbors and min-distance 0.1 over 10 trials. Because the Kozachenko–Leonenko estimator depends on k-NN distances, larger datasets yield artificially lower entropy due to smaller inter-point gaps. To draw fair comparisons, we sample $N$ points per dataset (without replacement) many times and average entropy across iterations to reduce variance.

For Leetcode vs. MBPP we use $N=150$ points per dataset over 250 trials ($k=4$), and for Codeforces $N=800$ over 250 trials ($k=21$) (Figure 2). "Diversity relative to X" is plotted as a difference for visualization only; diversity itself is a unary function. The results show that the entropy aligns with intuition: datasets expected to be more diverse exhibit higher entropy.

### 3.3 CHOOSING $k$ AND $d$

Kraskov et al. (2004) show that the bias–variance tradeoff for dataset size $N$ depends on $k/N$: larger $k$ increases bias but reduces variance and captures global structure. For KL-Divergence, we use $k \approx 4$ to emphasize local differences; for entropy, larger $k$ better captures global diversity, especially with scattered clusters; we find $k/N \in [0.02, 0.04]$ effective. Overall, diversity and entropy rankings are consistent across dimensions, and we recommend projecting to $d \in [8, 12]$ for dataset comparison. We present ablations for all hyper-parameters used to compute these metrics in Appendix A.

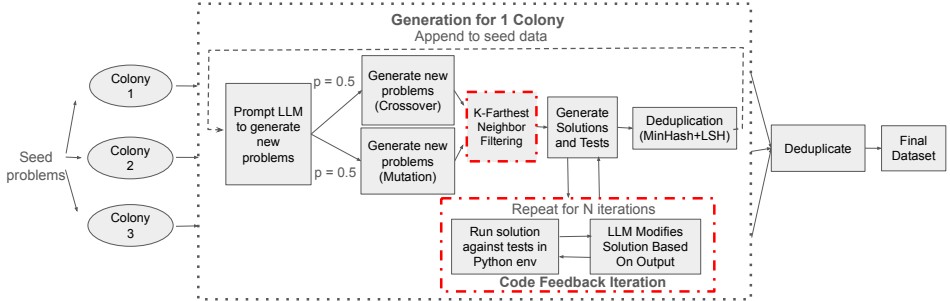

Figure 3: Generation Pipeline. Each colony receives a subset of the seed problems and applies mutation or crossover to them at each iteration. For each problem, it generates solutions and tests which go through multiple iterations of testing to ensure correctness. Deduplication removes similar problems within each colony; the remaining ones are used as seed data for the next iteration. The colony outputs are merged and deduplicated to produce the final dataset.

## 4 A Novel Benchmark Synthesis Pipeline

We introduce an end-to-end genetic algorithm for generating novel and diverse datasets from a seed dataset. The detailed algorithm is given in Appendix B, but we describe the main ideas here. Figure 3 provides a visual outline of the pipeline.

### 4.1 Data Generation Pipeline

**Mutation and Crossover.** Starting from the seed data, at each iteration, we randomly apply either crossover or mutation to generate new coding instructions (Majumdar et al., 2024). One key change from previous work is that our mutation prompts ask the model to modify an existing problem in three different difficulty variations: easier, equally difficult, and more difficult, encouraging diversity. We demonstrate the benefit of having multiple mutation difficulties in Section 5.2. Crossover prompts combine existing questions into new ones. Prompts are given in Appendix D.1, D.2. This example shows how mutation varies problem difficulty. Appendix F contains more examples.

---

**Seed Question:** Write a Python function to find the sum of an array.

**Hard Mutation Variant:** Write a Python function to find the sum of an array, where the array may contain nested lists of integers at any depth.

---

This example shows how crossover creates an interesting, novel question by combining two unrelated ones. Appendix F contains more examples.

---

**Seed Questions:** 1. Write a function to rotate a given list by a specified number of items to the right direction. 2. Write a function to find the maximum sum that can be formed which has no three consecutive elements present.

**Crossover Variant:** Write a function to rotate a list by a specified number of steps to the right, ensuring that the sum of any three consecutive elements in the newly rotated list does not exceed a given threshold.

---

**k-Farthest Neighbor Selection.** A key improvement of InfoSynth is that in order to increase novelty and diversity, we filter problems by cosine similarity to those already generated. In mutation, we produce easy, medium, and hard variants, retaining the two of three with lowest similarity to the seed and generated set. In crossover, we likewise generate three problems and keep the two least similar to the dataset.

**Iterative Code Feedback.** For each new problem, the model generates a Python solution and test cases (prompts in Appendix D.3, D.4). Candidate solutions are executed in an isolated environment, and the results are fed back to the model, which iteratively refines its solution and tests until all

Table 1: Dataset statistics and quality measures. Gen. Size: Initial generation size; Filtered: Problems removed via filtering; Avg. Tests: Avg. # test cases per problem; Human Correct: Human-verified correctness (%); Coverage: Test coverage (%); Hours: Person-hours spent generating.

| Dataset | Gen. Size | # Filtered | Avg. Tests | % Human Correct | % Coverage | Hours |
|---|---|---|---|---|---|---|
| MBPP-New | 1002 | 539 | 8.30 | 97% | 99% | 13 |
| MBPP-Guided | 992 | 572 | 8.86 | 98% | 99% | 14 |
| MBPP-Hard | 1007 | 223 | 10.35 | 96% | 100% | 14 |
| MBPP-Hard-Guided | 994 | 471 | 8.86 | 96% | 100% | 15 |
| Leetcode-New | 997 | 170 | 8.22 | 98% | 99% | 25 |
| Leetcode-Guided | 991 | 179 | 8.66 | 97% | 100% | 27 |

tests pass or a maximum number of iterations is reached. A key improvement of InfoSynth over prior methods is feeding the entire feedback history at each step, giving the model richer context; Section 5.3 explains how this induces chain-of-thought reasoning (Wei et al., 2022). Importantly, problems failing self-verification are excluded from the final dataset but still serve as seeds for the next generation round to encourage diversity.

**Deduplication.** We use the MinHash + LSH algorithm with 250 permutations and a 0.75 similarity threshold to remove textually similar problems, similar to that done by Majumdar et al. (2024).

**Postprocessing.** Generated problem descriptions are not always well-aligned with their test cases. For example, a problem may not describe how to handle edge-cases such as null inputs or empty arrays. In some problems, it is unreasonable to expect a test-taker to infer the desired behavior (e.g., should we return None or -1 on an empty array input?). An example of such a problem is given in Appendix E. For each problem-test pair, the model is prompted to rephrase the question to incorporate details on handling obscure edge-cases. The prompt is given in Appendix D.5.

### 4.2 EXPERIMENTAL SETUP

We generate six datasets using GPT-4o (Hurst et al., 2024) as the generator. The first dataset, called MBPP-Guided, is seeded with MBPP. The second dataset, MBPP-Hard-Guided, is also seeded with MBPP, but during mutation, the model is specifically prompted to make the questions more difficult. The third dataset, Leetcode-New, is seeded with a Leetcode dataset developed by Xia et al. (2025). For each of these three datasets, we perform the generation process again, this time without using K-farthest neighbor selection, resulting in a total of six datasets. These additional datasets are referred to as MBPP-New, MBPP-Hard, and Leetcode-New, respectively.

## 5 RESULTS AND ANALYSIS

We categorize generated problems as: (1) **Passing** (solution passes all tests), (2) **Failing** (fails $\geq$ 1 test), (3) **Erroring** (syntax/runtime error), and (4) **Unparsable** (malformed, e.g., missing [solution]/[test] tags, more common for smaller models). Table 1 reports benchmark statistics: test cases equal the number of `assert` statements, and test coverage is the fraction of code lines executed when all tests are run. We also evaluate SOTA models on all datasets (Table 2); Qwen2.5 models use 4-bit quantization. Since MBPP contains vague/misleading problems (Austin et al., 2021), we post-processed it for fairer comparison. Thus, Table 2 focuses on post-processed results as these provide the fairest comparison, omitting filtered versions (see Section 5.5). For each benchmark, we randomly sample 100 problems and manually verify that its solutions and test cases are fully correct and consistent with its problem statement.

### 5.1 NOVELTY AND DIVERSITY ANALYSIS

Figure 4a presents the novelty and diversity of MBPP-New and MBPP-Hard relative to MBPP-Original, while Figure 4b shows the same comparison for Leetcode-New relative to Leetcode-Original. Overall, our pipeline produces datasets that are more novel and diverse than the original seeds. However, filtering and post-processing reduce novelty compared to the initial generation. We attribute this to LLM memorization (Huang et al., 2025; Kiyomaru et al., 2024), since more novel

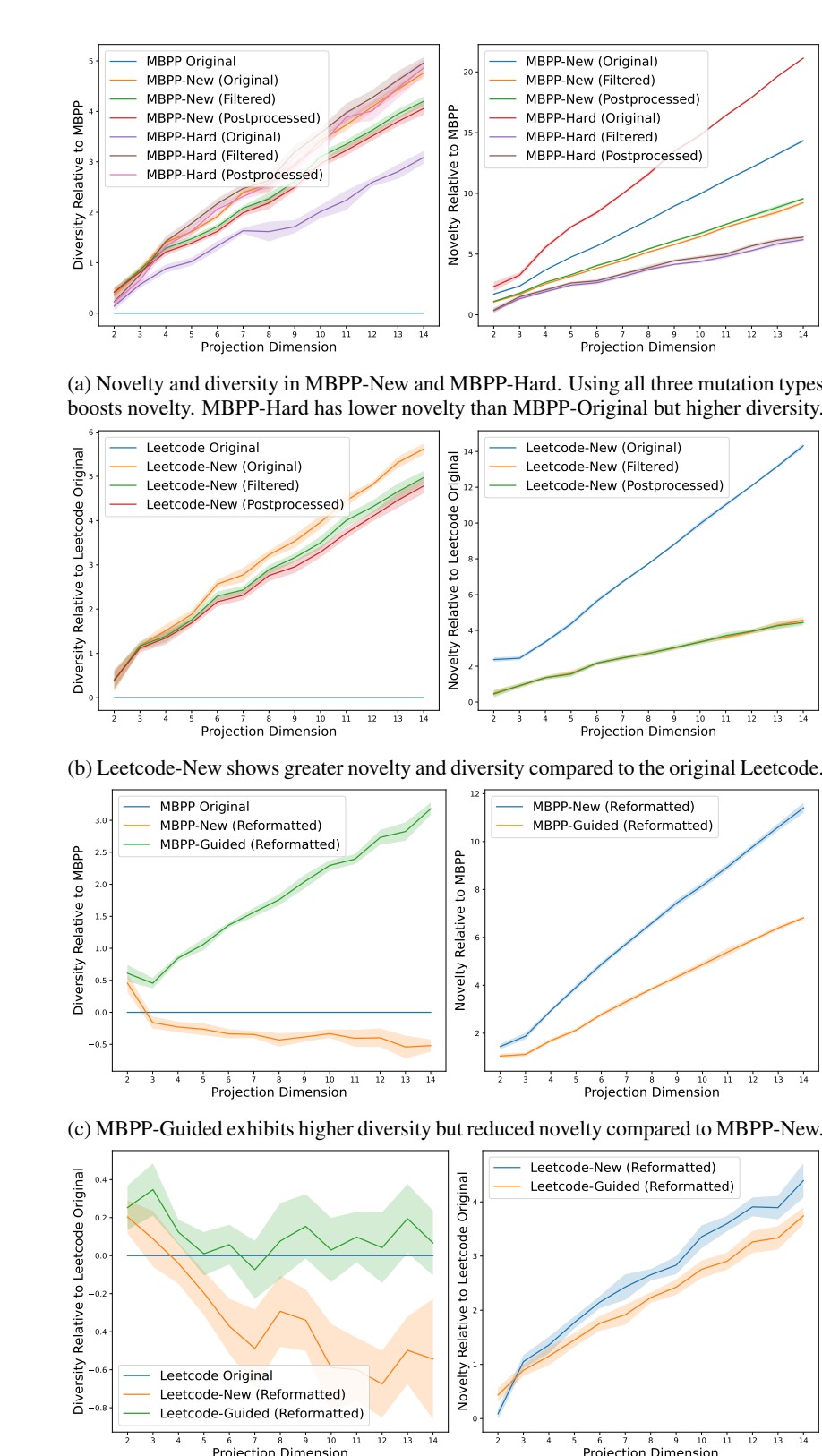

(a) Novelty and diversity in MBPP-New and MBPP-Hard. Using all three mutation types boosts novelty. MBPP-Hard has lower novelty than MBPP-Original but higher diversity.

(b) Leetcode-New shows greater novelty and diversity compared to the original Leetcode.

(c) MBPP-Guided exhibits higher diversity but reduced novelty compared to MBPP-New.

(d) Leetcode-Guided exhibits higher diversity but reduced novelty.

Figure 4: Novelty and diversity analysis across MBPP and Leetcode variants.

problems are often out-of-distribution and harder for the model to solve. All UMAP simulations use 80 neighbors and a minimum distance of 0.1, except for Figure 4d, which uses 30 neighbors due to smaller dataset size. In general, our results are not sensitive to UMAP hyperparameters.

Figures 4c and 4d show that k-farthest-neighbor filtering improves dataset novelty and diversity. This comes at the cost of generating easier problems (Table 2), highlighting another advantage of InfoSynth in controlling the novelty–diversity–difficulty tradeoff. Empirically, this arises because the generator struggles to produce difficult problems unless they are conceptually aligned with the seeds. We also observe a tradeoff between novelty and diversity: highly diverse datasets tend to concentrate around low-density regions in the seed-embedding distribution, which increases novelty but reduces diversity. Our results also show that filtering and post-processing reliably improve diversity.

## 5.2 EFFECT OF VARYING MUTATION DIFFICULTIES

Table 2 shows MBPP-Hard scores are 8%-15% lower than MBPP-Original across most models, suggesting hard mutations effectively raise difficulty. This comes with a tradeoff of the dataset having reduced diversity and novelty as the problems tend to be concentrated around fewer, but more challenging topics. Hence, the set of mutation difficulties can be carefully chosen in InfoSynth in order to control the difficulty of the produced benchmark.

## 5.3 EFFECT OF ITERATIVE CODE FEEDBACK

We find that passing solution-test pairs increase by 20% over 5 feedback iterations, showing the effectiveness of code iteration in producing robust problems. Error rates drop as the LLM fixes syntax/runtime issues, though the unparsable rate rises slightly due to occasional formatting failures. Appendix C shows feedback curves. Three iterations are typically ideal; further iterations yield marginal gains not worth the extra inference cost. We also find that iterative feedback acts as chain-of-thought (CoT) reasoning (Wei et al., 2022), as the model leverages the full feedback history to refine solutions/tests, lowering both error and failure rates. An example of this is in Appendix G.

## 5.4 CATEGORIZING FILTERED-OUT PROBLEMS

In general, we don't observe any major biases in the topics that make it past the filtering step, however, there are 2 notable types of problems that get filtered more often than others:

- Problems that are a crossover of problems that were already generated by crossover. This tends to create very difficult problems with many constraints, making it difficult for the generator LLM to generate accurate solutions and tests that adhere to every step. We interpret this as a limitation of current models' ability to perform long-horizon tasks, rather than a limitation of our pipeline.
- Problems involving significant numerical calculations (e.g. computing the nth Delannoy number); although the generator LLM often generates correct solutions for such problems, it struggles to make tests with accurate numerical assertions.

## 5.5 EFFECT OF THE POSTPROCESSING STEP

Appendix E shows two post-processed examples, where the model resolves ambiguous edge cases and sometimes rephrases statements more concisely without losing information. Table 2 shows 5–15% accuracy gains across most test-taker models, confirming that post-processing reduces ambiguity. Manual verification of 100 problems per dataset further shows 100% of post-processed problems are correctly reformatted without altering the core question.

## 5.6 RELATING DIVERSITY AND TOPIC COVERAGE

For each problem in MBPP Original, MBPP-New, and MBPP-Guided, we prompted GPT-4o-mini (Hurst et al., 2024) to provide up to 3 topic labels describing the problem, similar to the approach used by Zhao et al. (2025). The list of available topic labels was taken from the Leetcode dataset (kaysss, 2025). Figure 5 shows the results. The prompt is given in Appendix H.

Figure 5 shows that InfoSynth increases the number of problems that use each concept for most topics, creating more diverse and widely covering problems. We find that for some topics with lesser

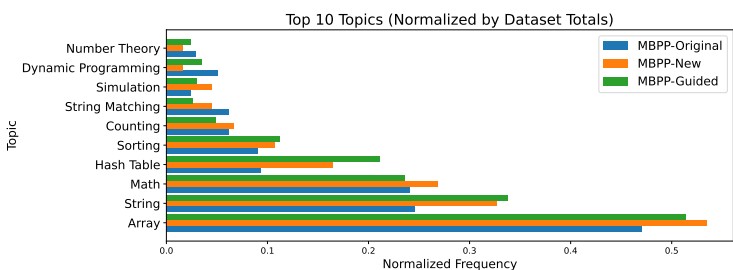

Figure 5: Fraction of problems relating to each topic for the 10 most common topics

coverage in the original MBPP dataset, our pipeline produces many more problems covering those topics. Evidently, MBPP-Guided has more topic coverage than MBPP-New across most categories, demonstrating the effectiveness of k-farthest-neighbor-filtering in encouraging diversity.

## 5.7 COMPARISON TO PREVIOUS GENERATION METHODS

We compare InfoSynth with GeneticInstruct and KodCode in generating novel and diverse problems. All three pipelines use Leetcode-based seeds, so novelty and diversity are measured against Leetcode Original. Since GeneticInstruct and KodCode use multiple seed datasets, we pool all reformatted problems generated by InfoSynth to form a combined dataset. Figure 6 shows the results.

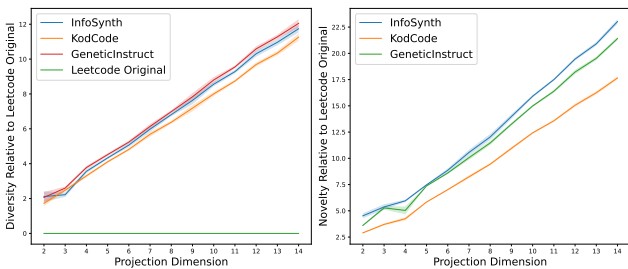

Figure 6: Novelty and diversity of various problem generation pipelines

We find that InfoSynth and GeneticInstruct exhibit very similar diversity, both slightly higher than KodCode. Additionally, InfoSynth stands out as the most novel dataset. Importantly, this comparison measures novelty against the Leetcode seed dataset *used by InfoSynth*, rather than the seed datasets for GeneticInstruct or KodCode, highlighting InfoSynth's superior ability to generate problems that differ substantially from its seed data.

## 5.8 CHOOSING AN EMBEDDING MODEL

We test various embeddings model on our datasets. Figures 7, 8 show that the relative novelty and diversity of datasets remains similar across embedding models despite some fluctuations in the magnitudes of those differences.

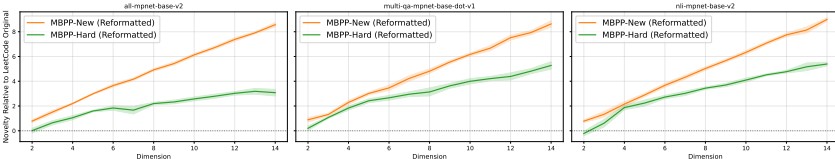

Figure 7: Novelty of datasets for various embedding models

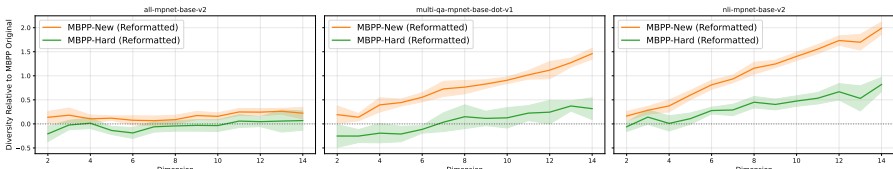

Figure 8: Diversity of datasets for various embedding models

Table 2: Test-taker performance on datasets

| Model | Dataset | Filtered | | | Postprocessed | | |
|---|---|---|---|---|---|---|---|
| | | %Pass | %Fail | %Err | %Pass | %Fail | %Err |
| Qwen2.5-7b-Instruct | MBPP-Original | 52.67 | 45.99 | 1.34 | 60.43 | 37.70 | 1.87 |
| | MBPP-New | 34.32 | 61.60 | 4.08 | 45.83 | 51.58 | 2.60 |
| | MBPP-Guided | - | - | - | 54.55 | 43.88 | 1.57 |
| | MBPP-Hard | 19.82 | 72.52 | 7.66 | 31.08 | 62.16 | 6.76 |
| | MBPP-Hard-Guided | - | - | - | 39.49 | 56.48 | 4.03 |
| | Leetcode-Original | - | - | - | 11.84 | 83.77 | 4.39 |
| | Leetcode-New | 25.29 | 74.12 | 0.59 | 25.88 | 72.35 | 1.76 |
| | Leetcode-Guided | - | - | - | 29.05 | 70.39 | 0.56 |
| Qwen2.5-3b-Coder | MBPP-Original | 46.79 | 48.93 | 4.28 | 52.41 | 42.51 | 5.08 |
| | MBPP-New | 31.35 | 55.47 | 13.17 | 38.78 | 49.54 | 11.69 |
| | MBPP-Guided | - | - | - | 40.56 | 49.13 | 10.31 |
| | MBPP-Hard | 21.62 | 54.95 | 23.42 | 24.77 | 53.15 | 22.07 |
| | MBPP-Hard-Guided | - | - | - | 29.30 | 57.75 | 12.95 |
| | Leetcode-Original | - | - | - | 2.63 | 85.53 | 11.84 |
| | Leetcode-New | 18.24 | 62.94 | 18.82 | 25.53 | 64.12 | 12.35 |
| | Leetcode-Guided | - | - | - | 15.08 | 75.42 | 9.50 |
| GPT-4.1-Mini | MBPP-Original | 58.02 | 36.10 | 5.88 | 66.04 | 30.48 | 0.00 |
| | MBPP-New | 56.96 | 40.63 | 2.41 | 67.35 | 29.68 | 2.97 |
| | MBPP-Guided | - | - | - | 72.03 | 22.38 | 5.59 |
| | MBPP-Hard | 44.14 | 50.45 | 5.41 | 55.86 | 38.29 | 5.86 |
| | MBPP-Hard-Guided | - | - | - | 68.58 | 27.60 | 3.82 |
| | Leetcode-Original | - | - | - | 32.89 | 54.82 | 12.28 |
| | Leetcode-New | 45.28 | 38.24 | 16.47 | 50.00 | 40.59 | 9.41 |
| | Leetcode-Guided | - | - | - | 48.60 | 46.37 | 5.03 |
| Gemini-2.0-Flash | MBPP-Original | 64.97 | 35.03 | 0.00 | 68.72 | 31.02 | 0.26 |
| | MBPP-New | 53.99 | 45.64 | 0.37 | 63.64 | 36.18 | 0.19 |
| | MBPP-Guided | - | - | - | 72.03 | 26.92 | 1.05 |
| | MBPP-Hard | 44.59 | 52.25 | 3.15 | 50.00 | 45.05 | 4.95 |
| | MBPP-Hard-Guided | - | - | - | 62.63 | 34.82 | 2.55 |
| | Leetcode-Original | - | - | - | 32.46 | 64.47 | 3.07 |
| | Leetcode-New | 44.71 | 54.12 | 1.18 | 51.18 | 46.47 | 2.35 |
| | Leetcode-Guided | - | - | - | 46.37 | 50.28 | 3.35 |
| Claude 3.7 Sonnet | MBPP-Original | 63.37 | 36.63 | 0.00 | 70.86 | 29.14 | 0.00 |
| | MBPP-New | 55.29 | 44.71 | 0.00 | 64.75 | 35.25 | 0.00 |
| | MBPP-Guided | - | - | - | 74.83 | 24.83 | 0.35 |
| | MBPP-Hard | 45.05 | 52.25 | 2.70 | 57.21 | 40.09 | 2.70 |
| | MBPP-Hard-Guided | - | - | - | 66.24 | 33.12 | 0.64 |
| | Leetcode-Original | - | - | - | 31.14 | 67.11 | 1.75 |
| | Leetcode-New | 44.71 | 54.71 | 0.59 | 44.71 | 55.29 | 0.00 |
| | Leetcode-Guided | - | - | - | 48.60 | 51.40 | 0.00 |
| o4-mini | MBPP-Original | 66.58 | 33.42 | 0.00 | 70.05 | 29.68 | 0.27 |
| | MBPP-New | 58.26 | 41.19 | 0.56 | 70.13 | 29.68 | 0.19 |
| | MBPP-Guided | - | - | - | 77.27 | 22.55 | 0.17 |
| | MBPP-Hard | 47.75 | 49.55 | 2.70 | 63.06 | 32.88 | 4.05 |
| | MBPP-Hard-Guided | - | - | - | 72.61 | 26.11 | 1.27 |
| | Leetcode-Original | - | - | - | 38.60 | 58.77 | 2.63 |
| | Leetcode-New | 40.59 | 58.82 | 0.59 | 50.59 | 47.06 | 2.35 |
| | Leetcode-Guided | - | - | - | 46.93 | 51.96 | 1.18 |

## 6 CONCLUSION

In this paper, we introduced InfoSynth, a novel framework to efficiently calculate the diversity and novelty of new benchmarks in a more-efficient and cost-effective manner. Moreover, we demonstrated the effectiveness of InfoSynth in generating high-quality, novel, and diverse synthetic coding datasets from seed data. We hope that future work will leverage our ideas to create robust, novel, and diverse benchmarks.

## 7 ETHICS STATEMENT

This work does not involve human subjects, personally identifiable data, or sensitive attributes. All datasets used (MBPP, LeetCode, and Codeforces) are publicly available, and our generated benchmarks were produced through synthetic problem generation and automated verification. We have carefully ensured that no private or proprietary code was included. Potential risks include the misuse of generated benchmarks for unfair evaluation or dataset contamination in future model training; to mitigate this, we document our pipeline in detail and encourage responsible use. We adhered to the ICLR Code of Ethics throughout the research and submission process.

## 8 REPRODUCIBILITY STATEMENT

We provide detailed descriptions of our dataset generation pipeline (Section 4), evaluation setup (Section 5), and algorithmic parameters (Appendix B). All prompts used for mutation, crossover, and verification are listed in Appendix D, and examples of generated problems are in Appendix F. Post-processing steps and deduplication methods are described in Section 4 and Appendix E. To facilitate reproducibility, we plan to release the full codebase and generated datasets upon publication.

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

## A  SENSITIVITY ANALYSIS FOR $k$ AND UMAP PARAMETERS

We present ablations for hyper-parameters used in our experiments. In general, our results generalize robustly across a wide range of hyper-parameters. Note that in the cases where changing a hyperparameter affects the novelty or diversity, it affects all datasets by roughly the same amount. Since our analysis looks at *relative* differences between benchmarks, our conclusions are still robust.

Specifically, we ablate three hyperparameters in Figure 9:

- $k$: the number of k-NN neighbors considered in the estimators
- n_neighbors: the number of neighbors used in the UMAP algorithm
- min_dist: the minimum distance used in the UMAP algorithm

## B  ALGORITHM FOR PROBLEM GENERATION

For MBPP-New we used $N = 1000$, $N_c = 10$, $B_s = 30$, $C = 2$, $B_c = 5$, $N_{it} = 5$. For MBPP-Hard we used $N = 500$, $N_c = 10$, $B_s = 15$, $C = 1$, $B_c = 4$, $N_{it} = 5$. For Leetcode-New we used $N = 1000$, $N_c = 10$, $B_s = 30$, $C = 2$, $B_c = 4$, $N_{it} = 5$.

For MBPP-Guided we used $N = 1000$, $N_c = 10$, $B_s = 30$, $C = 3$, $B_c = 5$, $N_{it} = 3$. For MBPP-Hard-Guided we used $N = 500$, $N_c = 10$, $B_s = 15$, $C = 3$, $B_c = 4$, $N_{it} = 3$. For Leetcode-Guided we used $N = 1000$, $N_c = 10$, $B_s = 30$, $C = 3$, $B_c = 4$, $N_{it} = 3$.

The algorithm is given in Algorithm 1

## C  CODE-FEEDBACK RESULTS

Figures 10, 11 show how the proportion of problems that pass, fail, error, and are unparsable changes as a function of the number of code feedback iterations. The results shown are consistent across all datasets; in general, more than 3 feedback iterations provides minimal gains in pass rate.

## D  PROBLEM GENERATION PIPELINE PROMPTS

We note that our prompts share some similarity with those used by Majumdar et al. (2024).

### D.1  MUTATION PROMPTS

```
% easy
Please decrease the difficulty of the given programming test
question a bit.
The new problem should be conceptually similar to the given
question,
but should not simply paraphrase it. Do not provide any hints,
solutions
or outputs. Only one new instruction is allowed.
Original Question: {instruction}
New Question:

% medium
Please create a new programming problem of the same difficulty as
the
given programming test question. The new problem should be
conceptually
similar to the given question, but should not simply paraphrase
it.
Do not provide any hints, solutions or outputs. Only one new
instruction is allowed.
```

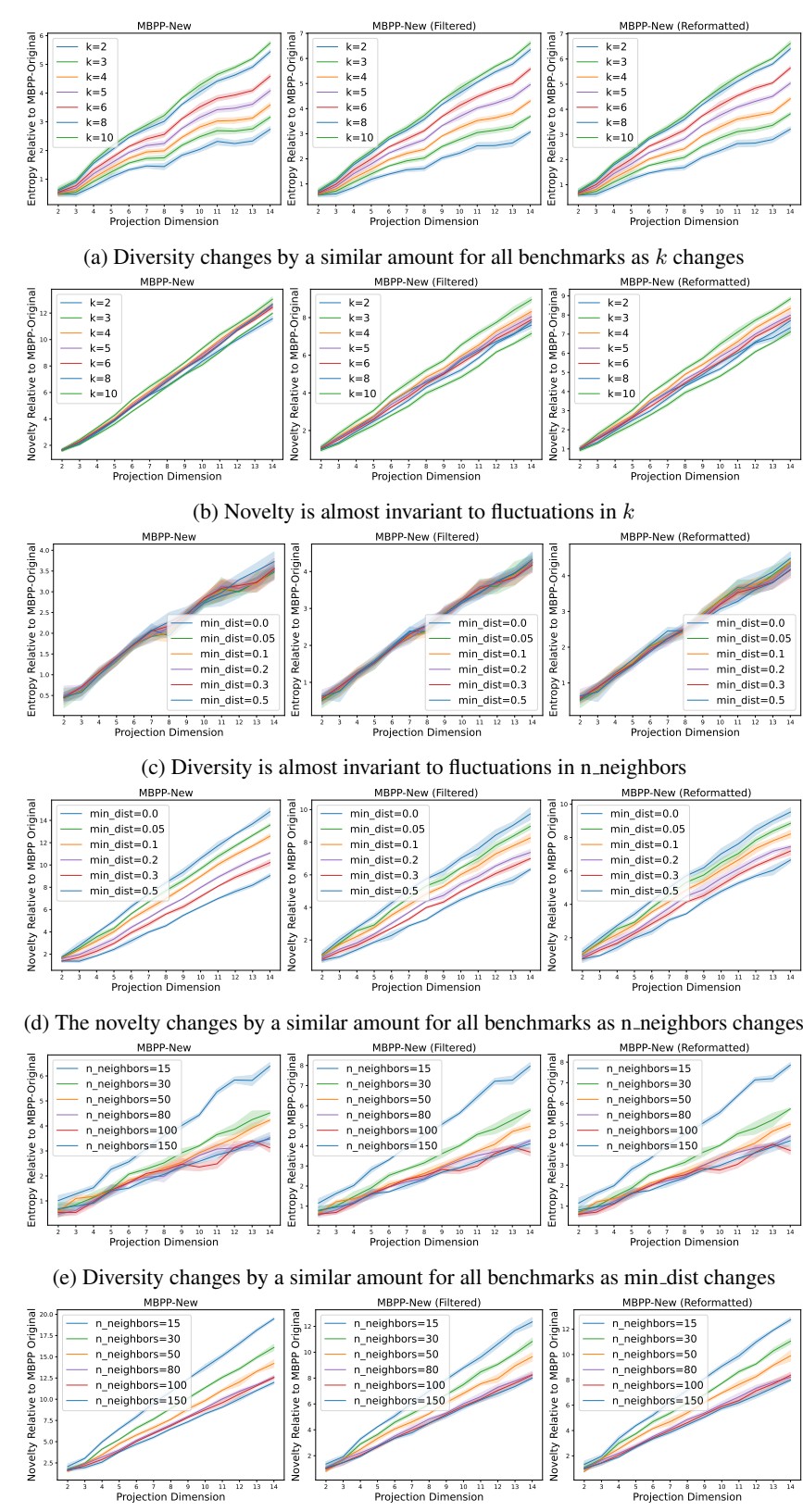

(a) Diversity changes by a similar amount for all benchmarks as $k$ changes

(b) Novelty is almost invariant to fluctuations in $k$

(c) Diversity is almost invariant to fluctuations in n_neighbors

(d) The novelty changes by a similar amount for all benchmarks as n_neighbors changes

(e) Diversity changes by a similar amount for all benchmarks as min_dist changes

(f) Novelty changes by a similar amount for all benchmarks as min_dist changes

Figure 9: Sensitivity analysis across all hyperparameters.

---

**Algorithm 1:** Problem Generation with Evolutionary Strategies

**Input:** $N$: total number of problems to generate

$N_c$: number of colonies

$B_s$: number of problems to sample as the seed data for each colony

$C$: number of problems to generate per crossover operation

$B_c$: crossover seed batch size; this is the number of problems fed into the model for it to combine

$N_{\text{it}}$: number of code feedback iterations

Generate(*seedData, numSamples*):
    **Initialize** problems $\leftarrow \emptyset$
    **for** *colony* $\leftarrow 1$ **to** $N_c$ **do**
        $N_s \leftarrow N/N_c$
        colonySeedData $\leftarrow$ Random sample of size $B_s$ from seedData
        problems $\leftarrow$ problems $\cup$ EvolveColony (colonySeedData, $N_s$)
        **deduplicate**(problems)
    **end**
    **return** *problems*

EvolveColony(*seedData, $N_s$*):
    newProblems $\leftarrow \emptyset$
    **while** |*newProblems*| $< N_s$ **do**
        operation $\leftarrow$ "mutation" with $p = 0.5$, "crossover" with $p = 0.5$
        **if** *operation == "mutation"* **then**
            problem $\leftarrow$ random sample of size 1 from seedData
            problems $\leftarrow$ problems $\cup$ **mutate**(problem)
        **else if** *operation == "crossover"* **then**
            batch $\leftarrow$ random sample of size $B_c$ from seedData
            problems $\leftarrow$ problems $\cup$ **crossover**($C$, batch)
        **end**
        **if** k-farthest-neighbor-filtering-enabled **then**
            $U \leftarrow$ newProblems $\cup$ seedData
            problems $\leftarrow$ select $K$ problems with **least cosine similarity** to $U$
        **end**
        newProblems $\leftarrow$ newProblems $\cup$ problems
        **deduplicate**(newProblems)
        seedData $\leftarrow$ seedData $\cup$ problems
        **deduplicate**(seedData)
    **end**
    **return** *newProblems*

GenerateSolutionsTests(*problem*):
    **GenerateTests**(problem)
    **GenerateSolutions**(problem)
    **for** $i = 1$ **to** $N_{it}$ **do**
        Run tests against solutions
        Feed output back into LLM to modify tests and solution
    **end**

---

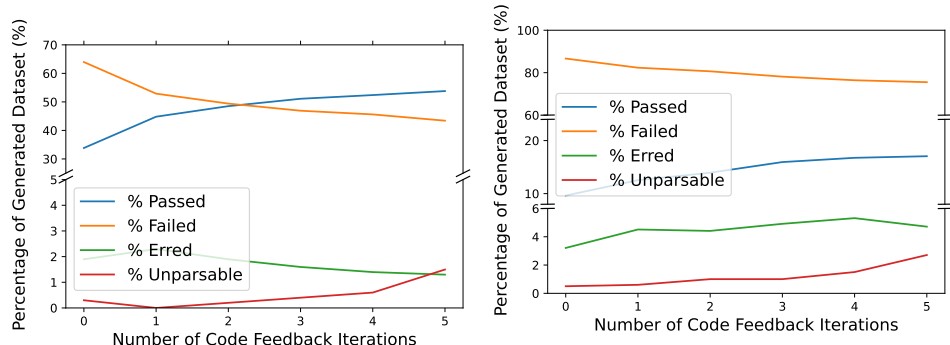

Figure 10: Self-Verification for MBPP-New

Figure 11: Self-Verification for Leetcode-New

```
Original Question: {instruction}
New Question:

% hard
Please increase the difficulty of the given programming test
question a bit.
Do not provide any hints, solutions or outputs. Only one new
instruction is allowed.
Original Question: {instruction}
New Question:
```

## D.2 CROSSOVER PROMPT

```
I will provide you with a set of coding questions. Please give me
a new coding
question that combines core concepts from two or more of the given
questions.
Please ensure that the new question is novel and does not simply
paraphrase
any of the problems I am giving you. Do not include any extra
information
that would help a test-taker other than the problem statement
itself.

Question 1:
{instruction 1}

Question 2:
{instruction 2}
...

New Question:
```

## D.3 SOLUTION & TEST GENERATION PROMPT

We note that our prompts are similar to those used by Xu et al. (2025).

```
You are an expert in Python coding.
## Task:
Please answer the question and generate unit tests to verify your
answer.
```

```
## Output Format:
Your solution and unit tests should be presented in the format
within the
specified sections below. Ensure your code is within code blocks.
For the
tests, use pytest style by defining individual test functions
(without
classes) and using assert statements. Your tests should be
implementation
independent. Ensure that you include the <|Solution Begin|>,
<|Solution End|>, <|Test Begin|>, and <|Test End|> tags as
depicted. The
solution function must be named solution.

<|Solution Begin|>
{Solution Code in Python}
<|Solution End|>

<|Test Begin|>
{Unit Test Code in Python}
<|Test End|>

## Example
Below is an example output format implementing a simple a + b
function.
<|Solution Begin|>
def add(a, b):
    \"\"\"Returns the sum of a and b.\"\"\"
    return a + b
<|Solution End|>

<|Test Begin|>
from solution import add
def test_add_positive_numbers():
    assert add(2, 3) == 5
def test_add_with_zero():
    assert add(0, 5) == 5
    assert add(5, 0) == 5
def test_add_negative_numbers():
    assert add(-1, -1) == -2
def test_add_mixed_sign_numbers():
    assert add(-1, 3) == 2
<|Test End|>

## Question:
{problem}
```

## D.4 SOLUTION & TEST GENERATION WITH ITERATIVE FEEDBACK PROMPT

```
You are an expert in Python coding.
## Task:
Please answer the question and generate unit tests to verify your
answer. The
entire chat history of your previous attempts to generate
questions and unit
tests is presented below in the "Chat History" section, along with
the output
of running your solution against your tests in a code execution
environment.
```

```
Please modify only your tests and/or solution to be more correct.

## Output Format:
<Same as "Output Format" section above>

## Chat History:
Attempt 1 Solution:
{attempt 1 solution}

Attempt 1 Code Execution Output:
{attempt 1 code output}

Attempt 2 Solution:
{Attempt 2 solution}

Attempt 2 Code Execution Output:
{attempt 2 code output}
...

## Question:
{problem}
```

### D.5 POSTPROCESSING PROMPTS

```
You are an expert in Python coding. Here is a coding problem
with associated test cases. Please rephrase the question so it
describes what the user should output for edge-cases without
changing the essence of the problem. Add as little information
as possible, only describing what the user should output for
edge-cases that cannot be inferred from the problem description.
Do not include anything except for the rewritten problem in your
response and do not include the test cases.
Question:
{question}
Tests:
{tests}
```

## E EXAMPLE OF POST-PROCESSED PROBLEMS

**MBPP-New Dataset Original Problem:**

```
Write a function that filters a list of usernames stored in a
dictionary, returning only those associated with students who fall
within a specified age range.
```

**Post-Processed Version:**

```
Write a function that filters a list of usernames stored in a
dictionary, returning only those associated with students who fall
within a specified age range. Ensure that the function returns an
empty list when there are no students or when the input dictionary
is empty.
```

**Leetcode-New Dataset Original Problem:**

```
Alice and Bob are engaged in a strategic game on an infinite 2D
plane with n points provided by their coordinates in two integer
arrays, xCoord and yCoord. They take turns, starting with Alice,
attacking a point on the plane to capture it, with the condition
```

that once a point is attacked, it is removed permanently from the
game, and they have to remove exactly 1 point per turn.

The winner is the player who either removes the point that leaves
no rectangle capable of being formed using the remaining points on
their turn or can force the scenario by optimal play such that the
opponent has no such move left on their subsequent turns.

Given the arrays xCoord and yCoord, along with knowledge of
optimal strategies for both players, determine if Alice, who
starts the game, can always guarantee a win. Return true if Alice
has a winning strategy, or false if Bob can always force a win
even with Alice starting first.

**Post-Processed Version:**

Alice and Bob are playing a strategic game on an infinite 2D
plane with n points defined by their coordinates in two integer
arrays, xCoord and yCoord. They alternately take turns, with Alice
starting first, to attack and permanently remove exactly one point
at a time. The objective is for a player to leave no possibility
of forming a rectangle using any four of the remaining points. A
player wins if they achieve this or if they can force a scenario
where the opponent has no such moves left. Given the arrays xCoord
and yCoord, determine if Alice has a guaranteed winning strategy.
Return true if Alice can always win, and false if Bob can always
force a win or if there are no points to start with.

# F    EXAMPLES OF GENERATED PROBLEMS

We note that for conciseness, the examples shown in this section are before the postprocessing
step. In this section, we illuminate the different mechanisms through which mutation and crossover
generate interesting problems.

## F.1    MUTATION OPERATION: ADDING OR REMOVING CONSTRAINTS

This example shows how mutation creates three variants of the question by adding requirements
or removing constraints from the problem. We can see that all three problems require similar
conceptual understanding, but the harder ones simply require more code or bookkeeping.

Original Question:
Write a function to filter the height and width of students which
are stored in a dictionary.

Easy Mutation:
Write a function that filters a list of usernames stored in a
dictionary, returning only those associated with students who fall
within a specified age range. Ensure that the function returns an
empty list when there are no students or when the input dictionary
is empty.

Medium Mutation:
Create a function that filters user profiles based on a
dictionary. The function should return a list of user IDs for
profiles where the age is within a specified range (inclusive)
and the profile only contains lowercase alphabetic characters. If
there are no valid profiles that match the criteria, the function
should return an empty list.

```
Hard Mutation:
Create a function to filter user passwords from a dictionary,
returning only those that are valid for students whose dimensions
(height, width, and weight) are within a given range. Each valid
password must include at least one uppercase letter, one lowercase
letter, one digit, and one special character. Ensure the solution
appropriately handles and returns an empty dictionary for cases
where no users are provided or all entries are invalid due to
dimension or password criteria.
```

## F.2 MUTATION OPERATION EXAMPLE: CREATIVE MODIFICATION

This example shows how mutation creates three variants of the question by creatively modifying the central idea of the problem itself. This differs from the previous example; in this case, the harder questions require a fundamental understanding of new topics.

```
Original Question:
Write a function to find the perimeter of a rectangle.

Easy Mutation:
Write a function to find the area of a rectangle.

Medium Mutation:
Write a function to calculate the area of a trapezoid given its
base lengths and height.

Hard Mutation:
Write a function to find the area of a rhombus given its diagonals
and verify if the rhombus is also a square by using its side
lengths.
```

## F.3 CROSSOVER OPERATION: COMBINING CONCEPTS

This example shows how crossover creates an interesting, novel question by combining two unrelated concepts. A key difference between crossover and mutation is that crossover does not introduce any new concepts or content into the generated problem as it draws from existing ones. This demonstrates the necessity of combining mutation and crossover into one pipeline; mutation introduces new concepts into the dataset while crossover takes existing concepts and uses them to create richer problems.

**Seed Problems:**

```
Question 1:
There is a 50 x 50 chessboard with one knight and some pawns on
it. You are given two integers kx and ky where (kx, ky) denotes
the position of the knight, and a 2D array positions where
positions[i] = [xi, yi] denotes the position of the pawns on the
chessboard.
Alice and Bob play a turn-based game, where Alice goes first. In
each player's turn:

The player selects a pawn that still exists on the board and
captures it with the knight in the fewest possible moves. Note
that the player can select any pawn, it might not be one that can
be captured in the least number of moves.
In the process of capturing the selected pawn, the knight may pass
other pawns without capturing them. Only the selected pawn can be
captured in this turn.
```

Alice is trying to maximize the sum of the number of moves made by
both players until there are no more pawns on the board, whereas
Bob tries to minimize them.
Return the maximum total number of moves made during the game that
Alice can achieve, assuming both players play optimally.
Note that in one move, a chess knight has eight possible positions
it can move to, as illustrated below. Each move is two cells in a
cardinal direction, then one cell in an orthogonal direction.

Question 2:
You are given an array points where points[i] = [xi, yi]
represents the coordinates of a point on an infinite plane.
Your task is to find the maximum area of a rectangle that:

Can be formed using four of these points as its corners.
Does not contain any other point inside or on its border.
Has its edges parallel to the axes.

Return the maximum area that you can obtain or -1 if no such
rectangle is possible.

**Crossover Generated Question:**

Alice and Bob are playing a strategic game on an infinite 2D
plane with n points defined by their coordinates in two integer
arrays, xCoord and yCoord. They alternately take turns, with Alice
starting first, to attack and permanently remove exactly one point
at a time. The objective is for a player to leave no possibility
of forming a rectangle using any four of the remaining points. A
player wins if they achieve this or if they can force a scenario
where the opponent has no such moves left. Given the arrays xCoord
and yCoord, determine if Alice has a guaranteed winning strategy.
Return true if Alice can always win, and false if Bob can always
force a win or if there are no points to start with.

F.4   CROSSOVER OPERATION: MULTISTEP REASONING

This example shows how crossover sometimes creates problems that require multistep reasoning.
The new problem effectively requires a test-taker to solve both the seed problems.

**Seed Problems:**

Question 1:
Write a python function to find the sum of fourth power of first n
odd natural numbers.

Question 2:
Write a python function to find the sum of xor of all pairs of
numbers in the given array.

**Crossover Generated Question:**

Write a python function to compute the combined metric of two
properties for a given array of integers:

1. The sum of the fourth power of the first 'k' odd numbers from
the array.

2. The sum of the bitwise XOR of all pairs of numbers formed by selecting one element from the first 'k' even numbers and one element from the last 'k' elements in the array.

The function should take two arguments – the array of integers and an integer 'k'. Ensure that 'k' is less than or equal to half the length of the array and that the array contains a sufficient number of odd and even numbers.

## G   EXAMPLE OF CHAIN-OF-THOUGHT THINKING

This example shows how the model is able to reason about why its solution and tests are erroring due to receiving feedback from the code execution environment.

**Question:** Write a function that takes a list of strings and returns the second shortest string that contains all vowels, sorted alphabetically. If no such string exists, return an empty string.

**Initial Solution:**

```
def solution(strings):
    """
    Returns the second shortest string containing all vowels,
sorted alphabetically.
    """
    all_vowels = set('aeiou')

    # Filter strings that contain all vowels
    filtered_strings = [s for s in strings if all_vowels <=
set(s)]

    # Sort them by length first and then alphabetically
    filtered_strings.sort(key=lambda x: (len(x), x))

    # Return the second shortest string or empty string if it
doesn't exist
    return filtered_strings[1] if len(filtered_strings) > 1 else
''
{Tests Omitted for Conciseness}
```

**Solution after First Feedback iteration:**

```
Let's revise the test cases to ensure they correctly test the
function's behavior and check the solution implementation
and its handling of edge cases. First, we'll make sure the
solution properly handles strings with varying cases (uppercase
and lowercase) and that it verifies the full vowel set in a
case-insensitive manner. Additionally, let's adjust the tests to
cover more scenarios and better reflect the conditions described
in the question. Here's the revised version of the solution and
unit tests:

{Solutions Omitted for Conciseness}
{Tests Omitted for Conciseness}
```

**Solution after Second Feedback iteration:**

```
We need to ensure that our solution and tests align correctly with
the task. Let's refine both the solution and the tests so that
they pass correctly according to the given problem statement. The
solution needs to ensure that we filter strings containing all
```

vowels in any case, sort them, and return the second shortest. The test cases should reflect this logic.

Here's the refined solution and tests:

{Solutions Omitted for Conciseness}
{Tests Omitted for Conciseness}

## H   TOPIC-LABELING PROMPT

You are a precise tagger for coding interview problems.

Given a problem statement and solution, choose up to THREE topics from the bank below that best describe the core techniques/data structures needed to solve it. Return JSON ONLY, no extra text, as:
{"topics": ["Topic1", "Topic2"]}

Rules:
- Use EXACT spellings from the bank (case/spacing must match).
- Prefer the most specific tag available (e.g., "Binary Tree" over "Tree", "Shortest Path" over "Graph" when appropriate).
- If the solution critically relies on a data structure (e.g., "Heap (Priority Queue)"), include it.
- If multiple techniques are essential (e.g., DP + Bitmask), include both.
- Do NOT exceed 3 topics; order them by importance.
- If nothing fits, choose the closest general tag (e.g., "Graph", "Array", "Math")|never invent tags.

Topic Bank (allowed values only):
Array; String; Hash Table; Dynamic Programming; Math; Sorting; Greedy; Depth-First Search; Binary Search; Database; Matrix; Tree; Breadth-First Search; Bit Manipulation; Two Pointers; Prefix Sum; Heap (Priority Queue); Simulation; Binary Tree; Graph; Stack; Counting; Sliding Window; Design; Enumeration; Backtracking; Union Find; Linked List; Number Theory; Ordered Set; Monotonic Stack; Segment Tree; Trie; Combinatorics; Bitmask; Divide and Conquer; Queue; Recursion; Geometry; Binary Indexed Tree; Memoization; Hash Function; Binary Search Tree; Shortest Path; String Matching; Topological Sort; Rolling Hash; Game Theory; Interactive; Data Stream; Monotonic Queue; Brainteaser; Doubly-Linked List; Randomized; Merge Sort; Counting Sort; Iterator; Concurrency; Probability and Statistics; Quickselect; Suffix Array; Line Sweep; Minimum Spanning Tree; Bucket Sort; Shell; Reservoir Sampling; Strongly Connected Component; Eulerian Circuit; Radix Sort; Rejection Sampling; Biconnected Component

Input:
[Problem]
{problem}

[Solution]
{solution}

```
Output:
JSON with key "topics" and UP TO 3 strings from the bank. No
prose, no explanations.
"""
```

