# OpenReview forum: "InfoSynth: Information-Guided Benchmark Synthesis for LLMs"
_ICLR.cc/2026/Conference — Submitted to ICLR 2026_

### Official Review · Reviewer_ZSEU · 2025-11-01

**Soundness:** 2
**Presentation:** 2
**Contribution:** 2
**Rating:** 4
**Confidence:** 4

**Summary:**

This paper proposed a novel framework for automatically generating and evaluting reasoning benchmarks guided by onformation-theoretic principles. Their experiments generate accurate test caes and solutions to new problems 97% of the time.

**Strengths:**

1. the research topic is interesting. While employing information-theoratic to evaluate the diversity of LLM's coding and resoning synthetsis is an interesting topic.
2. Overall, the paper is completed with a light presentation for each section. However, many typos need to be fixed.

**Weaknesses:**

1. The presentation is not good enough. For example, Line 092 "Schulman et al. uses" -> use. same for Line 093
2. The method lacks of sufficient novelty. Using the KL-divergence to estimate the gap between the current and the synthetic dataset. Hence, the metric of novelty and diversity are not impressive. Besides, your experiments don't show related section to discuss how these metrics are effective or not.

**Questions:**

N/A

---

> ### Author Response · Authors · 2025-11-21
> **Author Response to Reviewer ZSEU**
>
> We thank the reviewer for their insight and their time. We are grateful for the reviewer’s interest in our information-theoretic approach to efficiently quantifying the novelty and diversity of new benchmarks.
>
> **To address the first point**, we have updated the work to fix such typos and improved the overall presentation in the revised manuscript.
>
> **To address the second point: “The method lacks of sufficient novelty… your experiments don't show related section to discuss how these metrics are effective or not.”**
>
> Although KL-divergence is ubiquitously used to measure the distance between distributions, to our knowledge, we are the first to apply this intuition to benchmark analysis and demonstrate that it is meaningful. We have included some further experiments in section 3.2 (refer to Figures 1, 2) of the revised manuscript (changes in blue text) demonstrating that across benchmarks, our metrics are effective measures of novelty and diversity metrics.  We evaluate four datasets against each other; MBPP, Leetcode, HumanEval, and APPS [1, 2, 3, 4]. We find that Leetcode and APPS have high novelty compared to MBPP as their problems are harder and very different from MBPP, whereas HumanEval has low novelty as it is known to be more similar to MBPP. Similarly, we find that Leetcode and APPS have high diversity as their problems span many different computer science topics, whereas HumanEval has low diversity as the problems are easier and known to be more similar to MBPP. These results align with what we intuitively know to be true about these datasets.
>
> We greatly appreciate the reviewer’s constructive comments. We hope our response satisfies your concerns.
>
> References:
> 1. Jacob Austin, Augustus Odena, Maxwell I. Nye, Maarten Bosma, Henryk Michalewski, David Dohan, Ellen Jiang, Carrie J. Cai, Michael Terry, Quoc V. Le, and Charles Sutton. Program Synthesis with Large Language Models. ArXiv preprint, abs/2108.07732, 2021. URL https://arxiv.org/abs/2108.07732.
> 2. kaysss. Leetcode-problem-set. https://huggingface.co/datasets/kaysss/leetcode-problem-set, 2025. Accessed: 2025-05-11.
> 3. Mark Chen, Jerry Tworek, Heewoo Jun, Qiming Yuan, Henrique Ponde De Oliveira Pinto, Jared Kaplan, Harri Edwards, Yuri Burda, Nicholas Joseph, Greg Brockman, et al. Evaluating Large Language Models Trained on Code. ArXiv preprint, abs/2107.03374, 2021. URL https://arxiv.org/abs/2107.03374.
> 4. Dan Hendrycks, Steven Basart, Saurav Kadavath, Mantas Mazeika, Akul Arora, Ethan Guo, Collin Burns, Samir Puranik, Horace He, Dawn Song, and Jacob Steinhardt. Measuring coding challenge competence with apps. ArXiv preprint, abs/2105.09938, 2021. URL https://arxiv.org/abs/2105.09938.

---

> > ### Author Response · Authors · 2025-11-28
> >
> > Dear Reviewer ZSEU,
> >
> > With the December 2 rebuttal deadline approaching, we wanted to reach out to see whether you have any follow-up questions or clarifications regarding our submission. We would be glad to address any remaining concerns and provide further explanation where helpful. If possible, we would appreciate receiving any comments at your earliest convenience.
> > Thank you again for your time and thoughtful reviews.
> >
> > Best regards,
> >
> > The Authors

---

### Official Review · Reviewer_YFSU · 2025-11-01

**Soundness:** 2
**Presentation:** 3
**Contribution:** 3
**Rating:** 4
**Confidence:** 3

**Summary:**

This paper introduces InfoSynth, an information-theoretic framework for synthesizing benchmarks in code generation tasks. The key novelty lies in repurposing two classical information-theoretic metrics: (1) KL-Divergence for measuring benchmark novelty and (2) differential entropy for assessing internal diversity. These metrics provide an efficient way to quantify benchmark quality without costly model evaluations. Building on these metrics, the authors design an end-to-end pipeline that employs genetic algorithms and iterative code feedback to generate new Python coding problems, along with reference solutions and test cases, from seed datasets. The pipeline achieves 97% correctness in generating problems with accurate test cases and solutions.

**Strengths:**

1. The paper repurposes KL-Divergence and differential entropy to measure benchmark novelty and diversity. This is a creative contribution to the benchmark synthesis domain.

2. Section 3 effectively introduces the desirable properties (novelty and diversity) and provides both theoretical foundation and empirical validation with 95% confidence intervals.

3. The combination of genetic algorithms, mutation/crossover operations, and iterative code feedback is well-designed.

**Weaknesses:**

1. Limited Metric Validation Scope. The KL-Divergence and entropy validations primarily compare Leetcode subsets or extract subsets from larger benchmarks. While these experiments demonstrate the metrics work in clear-cut cases, *validation on independent benchmarks with subtle differences is missing. For example, comparing HumanEval vs MBPP, or benchmarks differing in problem-solving approaches (greedy vs dynamic programming) would better demonstrate discriminative power.

2. Estimator Stability and Negative KL Values. Although KL-Divergence is theoretically nonnegative, the estimator produces negative values in Figure 1 (RHS). The paper attributes this to subset-superset comparisons but does not quantify how frequently this occurs or its magnitude. This raises concerns about estimator stability and practical implications for the optimization process that maximizes KL-Divergence.

3. Unclear Pipeline Innovation Relative to Prior Work
The paper does not clearly articulate what specific innovations the pipeline introduces beyond combining existing techniques. For instance, how does integrating information-theoretic metrics into genetic algorithm selection differ from or improve upon prior evolutionary benchmark synthesis approaches?

**Questions:**

1. could you provide validation results comparing independent benchmarks (e.g., HumanEval vs MBPP) rather than only subsets from the same benchmark? This would strengthen evidence that your metrics capture novelty/diversity beyond obvious domain differences.

2. could you quantify how frequently negative KL values occur and their magnitudes? Have you explored alternative estimators or bootstrapping to assess stability?

---

> ### Author Response · Authors · 2025-11-21
> **Author Response to Reviewer YFSU**
>
> We thank the reviewer for their insight and their time. We are grateful for the reviewer’s interest in our information-theoretic approach to quantifying the novelty and diversity of new benchmarks. We also appreciate the reviewer’s interest in our pipeline that combines some existing techniques from prior work with new ideas in a cohesive way; a key contribution of the work is that it also introduces a simple, but novel problem-selection step based on distances between embedding vectors to increase diversity and novelty.
>
> **Q1: “could you provide validation results comparing independent benchmarks (e.g., HumanEval vs MBPP) rather than only subsets from the same benchmark? This would strengthen evidence that your metrics capture novelty/diversity beyond obvious domain differences.”**
>
> We have included some further experiments in the revised manuscript (changes in blue) demonstrating that across benchmarks, our metrics are effective measures of novelty and diversity metrics in section 3.2 (refer to Figures 1, 2).  We evaluate four datasets against each other; MBPP, Leetcode, HumanEval, and APPS [1, 2, 3, 4]. We find that Leetcode and APPS have high novelty compared to MBPP as their problems are harder and very different from MBPP, whereas HumanEval has low novelty as it is known to be more similar to MBPP. Similarly, we find that Leetcode and APPS have high diversity as their problems span many different computer science topics, whereas HumanEval has low diversity as the problems are easier and known to be more similar to MBPP. These results align with what we intuitively know to be true about these datasets.
>
> **Q2: “could you quantify how frequently negative KL values occur and their magnitudes? Have you explored alternative estimators or bootstrapping to assess stability?”**
>
> We find that in practice, negative KL values only occur when one dataset is almost entirely a subset of another. This creates a degenerate case where one distribution is trivially denser in its support causing the estimator to become negative in regions excluding the smaller dataset; bootstrapping also does not remediate this. However, when making practically useful comparisons between different datasets (ex. comparing MPBB and Leetcode) we find that KL-values are always positive. In our experiments, we never found the KL to be negative for any of the datasets we compared, except for the singular contrived subset/superset example shown in the paper. Hence, our methodology is useful in practice for comparing datasets as demonstrated through the empirical results. While we have not thoroughly explored alternative estimators we leave this for future work that may try to expand our novel framework for benchmark analysis to other metrics.
>
> We greatly appreciate the reviewer’s constructive comments. We hope our response satisfies your concerns.
>
> References:
> 1. Jacob Austin, Augustus Odena, Maxwell I. Nye, Maarten Bosma, Henryk Michalewski, David Dohan, Ellen Jiang, Carrie J. Cai, Michael Terry, Quoc V. Le, and Charles Sutton. Program Synthesis with Large Language Models. ArXiv preprint, abs/2108.07732, 2021. URL https://arxiv.org/abs/2108.07732.
> 2. kaysss. Leetcode-problem-set. https://huggingface.co/datasets/kaysss/leetcode-problem-set, 2025. Accessed: 2025-05-11.
> 3. Mark Chen, Jerry Tworek, Heewoo Jun, Qiming Yuan, Henrique Ponde De Oliveira Pinto, Jared Kaplan, Harri Edwards, Yuri Burda, Nicholas Joseph, Greg Brockman, et al. Evaluating Large Language Models Trained on Code. ArXiv preprint, abs/2107.03374, 2021. URL https://arxiv.org/abs/2107.03374.
> 4. Dan Hendrycks, Steven Basart, Saurav Kadavath, Mantas Mazeika, Akul Arora, Ethan Guo, Collin Burns, Samir Puranik, Horace He, Dawn Song, and Jacob Steinhardt. Measuring coding challenge competence with apps. ArXiv preprint, abs/2105.09938, 2021. URL https://arxiv.org/abs/2105.09938.

---

> ### Author Response · Authors · 2025-11-28
>
> Dear Reviewer YFSU,
>
> With the December 2 rebuttal deadline approaching, we wanted to reach out to see whether you have any follow-up questions or clarifications regarding our submission. We would be glad to address any remaining concerns and provide further explanation where helpful. If possible, we would appreciate receiving any comments at your earliest convenience.
> Thank you again for your time and thoughtful reviews.
>
> Best regards,
>
> The Authors

---

### Official Review · Reviewer_dyvZ · 2025-11-03

**Soundness:** 2
**Presentation:** 3
**Contribution:** 1
**Rating:** 2
**Confidence:** 3

**Summary:**

This paper proposes a method to calculate the novelty of a dataset using the KL divergence of embeddings, a method to calculate the diversity of a dataset using the entropy of embeddings, and a method for synthesizing coding problems based on genetic algorithms.

**Strengths:**

1. The experiment covers a wide range of LLMs, including open-weight models like Qwen, and SOTA models like GPT and Claude.
2. The implementation details are presented in detail. The prompt is also provided, so I think the reproducibility is good.
3. This paper is generally well-written and easy to follow.

**Weaknesses:**

1. **Whether KL can reflect the novelty of a dataset is questionable.** First, this paper calculates the KL Divergence of the problem statements in the dataset based on the embeddings output by a very small model. I believe this can at best only measure the differences between problem statements at the literal level, rather than truly capturing the novelty of these problems in their essence. For instance, if I add a large section of irrelevant background stories to each problem in a dataset, the KL Divergence can be significantly increased, yet the problems themselves remain unchanged in their essence. I believe using KL divergence to measure a certain type of "difference" and entropy to measure a certain type of "diversity" is valid. However, whether this approach can be effective ultimately depends on whether the projected distribution itself can establish a connection with the concepts such as "novelty" and "diversity" as perceived by the authors. This paper, however, have not provided sufficient validations in this regard.
2. **The correctness of synthetic problems cannot be guaranteed.** Both the test cases and solutions for these generated questions are produced by LLMs, and this paper only seems to ensure that the generated solutions and test cases are self-consistent, rather than ensuring both are correct. It is possible that both the test cases and solutions are erroneous in the same way.

**Questions:**

Please refer to "Weaknesses".

---

> ### Author Response · Authors · 2025-11-21
> **Author Response to Reviewer dyvZ**
>
> We thank the reviewer for their insight and their time. We are grateful for the reviewer’s interest in our information-theoretic approach to quantifying the novelty and diversity of new benchmarks, and our novel problem-generation pipeline along with our evaluation of it on a variety of test-taker LLMs.
>
> **To address the first concern: “Whether KL can reflect the novelty of a dataset is questionable.”**
>
> We agree that the KL cannot perfectly capture the semantics of the problem statement. However, we have included some further experiments in section 3.2 (refer to Figures 1, 2) of the revised manuscript (changes in blue text) demonstrating that across benchmarks, these metrics are still effective measures of novelty and diversity. We evaluate four datasets against each other; MBPP, Leetcode, HumanEval, and APPS [1, 2, 3, 4]. We find that Leetcode and APPS have high novelty compared to MBPP as their problems are harder and very different from MBPP, whereas HumanEval has low novelty as it is known to be more similar to MBPP. Similarly, we find that Leetcode and APPS have high diversity as their problems span many different computer science topics, whereas HumanEval has low diversity as the problems are easier and known to be more similar to MBPP. These results align with what we intuitively know to be true about these datasets.
>
> **To address the second concern: “The correctness of synthetic problems cannot be guaranteed.”**
>
> We want to highlight that the generated problems were **manually verified by humans** who checked that both tests and solutions were mutually consistent with the problem statement. While it is true that an LLM could generate solutions and tests that are merely self-consistent, we empirically find (through human-verification) that 97% of generated problems are fully correct so these failure cases do not happen very often.
>
> We greatly appreciate the reviewer’s constructive comments. We hope our response satisfies your concerns.
>
>
> References:
> 1. Jacob Austin, Augustus Odena, Maxwell I. Nye, Maarten Bosma, Henryk Michalewski, David Dohan, Ellen Jiang, Carrie J. Cai, Michael Terry, Quoc V. Le, and Charles Sutton. Program Synthesis with Large Language Models. ArXiv preprint, abs/2108.07732, 2021. URL https://arxiv.org/abs/2108.07732.
> 2. kaysss. Leetcode-problem-set. https://huggingface.co/datasets/kaysss/leetcode-problem-set, 2025. Accessed: 2025-05-11.
> 3. Mark Chen, Jerry Tworek, Heewoo Jun, Qiming Yuan, Henrique Ponde De Oliveira Pinto, Jared Kaplan, Harri Edwards, Yuri Burda, Nicholas Joseph, Greg Brockman, et al. Evaluating Large Language Models Trained on Code. ArXiv preprint, abs/2107.03374, 2021. URL https://arxiv.org/abs/2107.03374.
> 4. Dan Hendrycks, Steven Basart, Saurav Kadavath, Mantas Mazeika, Akul Arora, Ethan Guo, Collin Burns, Samir Puranik, Horace He, Dawn Song, and Jacob Steinhardt. Measuring coding challenge competence with apps. ArXiv preprint, abs/2105.09938, 2021. URL https://arxiv.org/abs/2105.09938.

---

> > ### Author Response · Authors · 2025-11-28
> >
> > Dear Reviewer dyvZ,
> >
> > With the December 2 rebuttal deadline approaching, we wanted to reach out to see whether you have any follow-up questions or clarifications regarding our submission. We would be glad to address any remaining concerns and provide further explanation where helpful. If possible, we would appreciate receiving any comments at your earliest convenience.
> > Thank you again for your time and thoughtful reviews.
> >
> > Best regards,
> >
> > The Authors

---

### Official Review · Reviewer_oqPV · 2025-11-04

**Soundness:** 3
**Presentation:** 3
**Contribution:** 3
**Rating:** 6
**Confidence:** 4

**Summary:**

This paper introduces a framework for automatically generating Python coding benchmarks for LLMs. The works makes two main contributions:
1. an information theoretic framework using KL-divergence and entropy to quantify benchmark novelty and diversity without expensive model evaluations
2. and end-to-end pipeline using genetic algorithms with iterative code feedback to synthesize verifiable problems.
The experiments show the correctness of newly generated problems achieves 97% and consistently exhibit higher novelty and diversity compared to seed datasets.

**Strengths:**

1. KL-divergence for novelty and entropy for diversity measure make sense, and are computationally efficient compared to model based evaluations.
2. This work includes several important ablation studies that validate key design choices:
- multiple mutation difficulties provide difficulty control
- iterative feedback substantially improves correctness
- postprocessing improves clarity
- k-farthest neighbor filtering increases diversity

**Weaknesses:**

1. The paper sets the goal of creating benchmarks that are contamination-free, while lacking the experiments or discussion of contamination evaluation on the proposed data generation method.
2. No comparison with other synthetic benchmark generation methods. Section 2 actually listed multiple comparable methods, but none of them were benchmarked in empirical study. Ideally the authors should show, with different dataset generation methods, compare novelty / diversity metrics, correctness rates, model performance distributions, computational costs.
3. Note that Table 1 shows this generation pipeline has high filtering rate, for example, for MBPP-Guided, more than half of the generated samples were discarded. The authors should provide careful analysis of the breakdown of filtering reasons, characterization of filtered problems, potential bias from filtering.

**Questions:**

1. How sensitive are the novelty/diversity metrics to the choice of embedding model, have you tested with other embeddings?
2. Can you provide evidence that generated benchmarks actually help with contamination detection or out-of-distribution evaluation in practice?
3. Can you provide analysis of the filtered problems, what types of problems got filtered and would this introduce systematic biases?
4. Can you provide sensitivity analysis on hyperparameter selection for k and UMAP parameters?

---

> ### Author Response · Authors · 2025-11-21
> **Author Response to Reviewer oqPV**
>
> We thank the reviewer for their insight and their time. We are grateful for the reviewer’s interest in our information-theoretic approach to quantifying the novelty and diversity of new benchmarks in a computationally-efficient manner.
>
> **To address the reviewer’s comment about "comparison with other synthetic benchmark generation methods:**
>
> In section 5.7 of the revised manuscript (changes in blue text) we have included a comparison of InfoSynth against KodCode and GeneticInstruct, two pipelines from prior works based on related ideas. To summarize, we find that InfoSynth and GeneticInstruct exhibit very similar diversity, both slightly higher than KodCode. Additionally, InfoSynth stands out as the most novel dataset. Importantly, this comparison measures novelty against the Leetcode seed dataset used by InfoSynth, rather than the seed datasets for GeneticInstruct or KodCode, highlighting InfoSynth’s superior ability to generate problems that differ substantially from its seed data.
>
> **Q1: How sensitive are the novelty/diversity metrics to the choice of embedding model, have you tested with other embeddings?**
>
> In section 5.8 of the revised manuscript, we have added experiments showing the sensitivity of novelty and diversity to choice of embedding model. To summarize, we find that novelty remains mostly unchanged. While there is some fluctuation in the magnitude of diversity, the relative rankings between datasets remains the same.
>
>
> **Q2: Can you provide evidence that generated benchmarks actually help with contamination detection or out-of-distribution evaluation in practice?**
>
> We believe that the performance of test-taker models on our datasets demonstrates their ability to aid in out-of-distribution evaluation. Test taker results on many of our datasets, notably MBPP-New and MBPP-Hard are lower than the original MBPP dataset, providing evidence that our datasets are different from the existing training data these models are often trained on. As described in the introduction, this work does not completely solve the task of creating contamination-free benchmarks, however, it provides an improved method of evaluating and generating benchmarks for novelty and diversity.
>
>
> **Q3: Can you provide analysis of the filtered problems, what types of problems got filtered and would this introduce systematic biases?**
>
> We have included some analysis in the revised manuscript (changes in blue) in section 5.4. In general, we don’t observe any major biases in the topics that make it past the filtering step, however, there are 2 notable types of problems that get filtered more often than others.
> 1. Problems that are a crossover of problems that were already generated by crossover. This tends to create very difficult problems with many constraints, making it difficult for the generator LLM to generate accurate solutions and tests that adhere to every step. We interpret this as a limitation of current models' ability to perform long-horizon tasks, rather than a limitation of our pipeline.
> 2. Problems involving significant numerical calculations (e.g. computing the nth Delannoy number); although the generator LLM often generates correct solutions for such problems, it struggles to make tests with accurate numerical assertions. We believe that future work may be able to remediate this by using stronger generator models or providing generator models with some simple computational tools.
>
>
> **Q4: Can you provide sensitivity analysis on hyperparameter selection for k and UMAP parameters?**
>
> We have included a hyperparameter sensitivity analysis in the revised manuscript in Appendix A. To summarize the results, we observe that novelty and diversity are often invariant to changes in some hyperparameters. Note that in the cases where changing a hyperparameter affects the novelty or diversity, it affects all datasets by roughly the same amount. Since our analysis looks at relative differences between benchmarks, our conclusions are still robust.

---

> ### Author Response · Authors · 2025-11-28
>
> Dear Reviewer oqPV,
>
> With the December 2 rebuttal deadline approaching, we wanted to reach out to see whether you have any follow-up questions or clarifications regarding our submission. We would be glad to address any remaining concerns and provide further explanation where helpful. If possible, we would appreciate receiving any comments at your earliest convenience.
> Thank you again for your time and thoughtful reviews.
>
> Best regards,
>
> The Authors

---

### Author Response · Authors · 2025-11-30
**Summary of changes and additional experiments during rebuttal**

We are grateful to all of the reviewers for their feedback. During the rebuttal period we substantially revised the paper (changes in blue text) and conducted additional experiments to address the main concerns, which we summarize here:

**Comparisons showing how InfoSynth affects novelty and diversity compared to prior work**

In section 5.7 of the revised paper we have included a comparison of InfoSynth against KodCode and GeneticInstruct, two pipelines from prior works based on related ideas. To summarize, we find that InfoSynth and GeneticInstruct exhibit very similar diversity, both slightly higher than KodCode. Additionally, InfoSynth stands out as the most novel dataset. Importantly, this comparison measures novelty against the Leetcode seed dataset used by InfoSynth, rather than the seed datasets for GeneticInstruct or KodCode, highlighting InfoSynth’s superior ability to generate problems that differ substantially from its seed data.

**Further experiments demonstrating that the KL-divergence and entropy are meaningful characterizations of novelty and diversity**

We have included further experiments in the revised paper demonstrating that across benchmarks, our metrics are effective measures of novelty and diversity metrics in section 3.2 (refer to Figures 1, 2). We evaluate four datasets against each other; MBPP, Leetcode, HumanEval, and APPS [1, 2, 3, 4]. We find that Leetcode and APPS have high novelty compared to MBPP as their problems are harder and very different from MBPP, whereas HumanEval has low novelty as it is known to be more similar to MBPP. Similarly, we find that Leetcode and APPS have high diversity as their problems span many different computer science topics, whereas HumanEval has low diversity as the problems are easier and known to be more similar to MBPP. These results align with what we intuitively know to be true about these datasets.

**Ablations demonstrating how choice of embedding model affects novelty and diversity**

In section 5.8 of the revised paper, we have added experiments showing the sensitivity of novelty and diversity to choice of embedding model. To summarize, we find that novelty remains mostly unchanged. While there is some fluctuation in the magnitude of diversity, the relative rankings between datasets remains the same.

**Ablations on the different hyperparameters used for computing novelty and diversity metrics**

We have included a hyper-parameter sensitivity analysis in Appendix A of the revised paper. To summarize, we observe that novelty and diversity are often invariant to changes in some hyperparameters. Note that in the cases where changing a hyperparameter affects the novelty or diversity, it affects all datasets by roughly the same amount. Since our analysis looks at relative differences between benchmarks, our conclusions are still robust.

**An analysis of the types of problems filtered out during the filtering step**

We have included analysis in the revised paper in section 5.4. In general, we don’t observe any major biases in the topics that make it past the filtering step, however, there are 2 notable types of problems that get filtered more often than others.
1. Problems that are a crossover of problems that were already generated by crossover. This tends to create very difficult problems with many constraints, making it difficult for the generator LLM to generate accurate solutions and tests that adhere to every step. We interpret this as a limitation of current models' ability to perform long-horizon tasks, rather than a limitation of our pipeline.
2. Problems involving significant numerical calculations (e.g. computing the nth Delannoy number); although the generator LLM often generates correct solutions for such problems, it struggles to make tests with accurate numerical assertions. We believe that future work may be able to remediate this by using stronger generator models or providing generator models with some simple computational tools.


References:

1. Jacob Austin, Augustus Odena, Maxwell I. Nye, Maarten Bosma, Henryk Michalewski, David Dohan, Ellen Jiang, Carrie J. Cai, Michael Terry, Quoc V. Le, and Charles Sutton. Program Synthesis with Large Language Models. ArXiv preprint, abs/2108.07732, 2021. URL https://arxiv.org/abs/2108.07732.
2. kaysss. Leetcode-problem-set. https://huggingface.co/datasets/kaysss/leetcode-problem-set, 2025. Accessed: 2025-05-11.
3. Mark Chen, Jerry Tworek, Heewoo Jun, Qiming Yuan, Henrique Ponde De Oliveira Pinto, Jared Kaplan, Harri Edwards, Yuri Burda, Nicholas Joseph, Greg Brockman, et al. Evaluating Large Language Models Trained on Code. ArXiv preprint, abs/2107.03374, 2021. URL https://arxiv.org/abs/2107.03374.
4. Dan Hendrycks, Steven Basart, Saurav Kadavath, Mantas Mazeika, Akul Arora, Ethan Guo, Collin Burns, Samir Puranik, Horace He, Dawn Song, and Jacob Steinhardt. Measuring coding challenge competence with apps. ArXiv preprint, abs/2105.09938, 2021. URL https://arxiv.org/abs/2105.09938.

---

### Meta-Review · Area_Chair_wEsU · 2025-12-26

**Summary:**

This paper introduces metrics based on KL divergence and entropy to quantify the novelty and diversity of benchmarks, and proposes a framework called InfoSynth, which automatically generates Python coding benchmarks for LLM using genetic algorithms and iterative code feedback on seed datasets. The research topic of benchmark generation is interesting, and the proposed framework is reasonable and easy to understand. However, this paper has the following main issues: 1) The technical innovation of InfoSynth is limited. Besides incorporating existing technologies, it is unclear what specific innovations the process introduces. 2) The connection between the KL divergence metric and the novelty of the benchmarks claimed by the authors lacks sufficient verification. 3) The quality of the generated datasets cannot be adequately guaranteed. The datasets generated by LLM may contain various errors and noise, while only 100 samples in each dataset were manually checked. 4) Comparison with other dataset construction methods is insufficient. In all, the work presented in this paper needs substantial improvement.

**Reviewer Concerns:**

Some minor concerns were addressed by the rebuttal, but the concerns mentioned in the above summary were not fully addressed.

**Reviewer Scores:**

Reviewer dyvZ would have increased the score slightly.

---

### Decision · Program_Chairs · 2026-01-26

Reject